# Legumes of the Sardinia Island: Knowledge on Symbiotic and Endophytic Bacteria and Interactive Software Tool for Plant Species Determination

**DOI:** 10.3390/plants11111521

**Published:** 2022-06-06

**Authors:** Rosella Muresu, Andrea Porceddu, Giuseppe Concheri, Piergiorgio Stevanato, Andrea Squartini

**Affiliations:** 1Institute for the Animal Production System in Mediterranean Environment-National Research Council (ISPAAM CNR), Traversa La Crucca 3, 07100 Sassari, SS, Italy; ro.muresu@tiscali.it; 2Department of Agriculture, University of Sassari, Viale Italia 1, 07100 Sassari, SS, Italy; aporceddu@uniss.it; 3Department of Agronomy, Food, Natural Resources, Animals and Environment, DAFNAE, University of Padova, Viale dell’Università 16, 35020 Legnaro, PD, Italy; giuseppe.concheri@unipd.it (G.C.); stevanato@unipd.it (P.S.)

**Keywords:** Sardinia, legumes, symbionts, nitrogen fixation, endophytes

## Abstract

A meta-analysis was carried out on published literature covering the topic of interactive plant microbiology for botanical species of legumes occurring within the boundary of the Italian island Sardinia, lying between the Tyrrhenian and the western Mediterranean seas. Reports were screened for the description of three types of bacterial occurrences; namely, (a) the nitrogen-fixing symbionts dwelling in root nodules; (b) other bacteria co-hosted in nodules but having the ancillary nature of endophytes; (c) other endophytes isolated from different non-nodular portions of the legume plants. For 105 plant species or subspecies, over a total of 290 valid taxonomical descriptions of bacteria belonging to either one or more of these three categories were found, yielding 85 taxa of symbionts, 142 taxa of endophytes in nodules, and 33 in other plant parts. The most frequent cases were within the *Medicago*, *Trifolium*, *Lotus*, *Phaseolus,* and *Vicia* genera, the majority of symbionts belonged to the *Rhizobium*, *Mesorhizobium*, *Bradyrhizobium,* and *Sinorhizobium* taxa. Both nodular and extra-nodular endophytes were highly represented by Gammaproteobacteria (*Pseudomonas*, *Enterobacter*, *Pantoea*) and Firmicutes (*Bacillus*, *Paenibacillus*), along with a surprisingly high diversity of the Actinobacteria genus *Micromonospora*. The most plant-promiscuous bacteria were *Sinorhizobium meliloti* as symbiont and *Bacillus megaterium* as endophyte. In addition to the microbial analyses we introduce a practical user-friendly software tool for plant taxonomy determination working in a Microsoft Excel spreadsheet that we have purposely elaborated for the classification of legume species of Sardinia. Its principle is based on subtractive keys that progressively filter off the plants that do not comply with the observed features, eventually leaving only the name of the specimen under examination.

## 1. Introduction

A botanical world census, although subjected to rapid changes and updates estimated that, excluding algae, mosses, liverworts, and hornworts, about 390,900 plants are known to science, of which approximately 369,400 are flowering [1]. Within the flowering angiosperms, the Leguminosae (Fabaceae) family encompasses, under conservative records, at least 19,000 known species within 751 genera, which constitute about 7% of the flowering plant species [2,3]. The Italian flora features 7672 species listed between printed and digital archives [4,5]. Within these, the taxa belonging to the legume family are indicated as 519, inclusive of subspecies, a large deal of which, namely 290, grow in the Sardinia island, located in an ecologically and climatically important position within the Mediterranean sea. The Sardinian legumes include also 27 endemics mostly within the Genista and Astragalus genera [6].

The Fabaceae are also the main plant family engaging in symbiotic relationships with nitrogen-fixing bacteria, broadly referred to as rhizobia, that induce the neo-organogenesis of nodules, mostly on the roots, inside which the bacteria carry out their mutualism [7,8].

The Leguminosae family is divided into six subfamilies: Cercidoideae, Detarioideae, Duparquetioideae, Dialioideae, Caesalpinioideae (which includes the former Mimosoideae), and Faboideae (Papilionoideae). The first four have relatively few species and those are not reported to form nodules, while the last two are the most species-rich and the ones that encompass nodulating species, particularly in the case of the Faboideae [9].

The legumes, similar to many other plants, release compounds such as flavonoids from their growing roots. For the rhizobia, these compounds have a particular meaning, which is that of specific inducers of the expression of a group of genes, the so-called *nod* genes. The activity of flavonoids depends on the concentration; at 10^−9^ M, away from the source of emission, they act as chemo-attractants, while at 10^−6^ M, close to the root, they induce the *nod* genes. The activation of these bacterial genes leads to the synthesis of the key signal, a molecule that, traveling from the bacterium to the plant, influences the behavior of the latter, forcing it to reprogram its root morphogenesis by triggering meristematic activities that will give rise to the formation of the nodule where the bacteria will be housed. A true neoplasm is triggered by a chitin-based compound with a fatty acid tail: a chito-lipo-oligo-saccharide [10]. For the bacterial world, chitin is by no means a common product, essentially belonging to fungi or arthropods, where it occurs, however, in much longer chains. The nod genes, present exclusively in the rhizobia, therefore, send a rather unusual signal, and with this they order the plant to build the rooms in which they will be housed. Among rhizobia and legumes, especially the herbaceous plants of temperate zones, there is also a strict host specificity, primarily exercised through the structure of the chito-lipo-oligo-saccharide. By way of example, *Rhizobium leguminosarum* biovar *viciae* nodulates the genera *Vicia*, *Pisum*, *Lathyrus*, and *Lens* as it produces a signal in which the first of the 4 or 5 chitin residues is acetylated and the fatty acid tail has a length of 18 atoms of carbon with 4 desaturations in precise positions. The different nucleotide sequence of the nod genes in the different rhizobia is responsible for the peculiarities of the structure of the signals of each species. It is interesting to note that by administering the purified compound to an alfalfa plant, even in the absence of rhizobia, that plant will specifically form nodules, obviously empty and not nitrogen-fixing, but structurally complete [11].

Concerning their overall microbiology, however, plants have been for a long time considered as organisms devoid of inner microbiota unless infected by specific pathogens or when hosting microsymbionts in well-confined organs, such as the above-mentioned rhizobia. Contrary to this belief, the concept of plant endophytism has been progressively gaining attention and its importance is now universally recognized [12]. The interaction between plants and their inner microorganisms is stirring interest also because the microbiome inside plants appears to present remarkable parallels with the intestinal microbiome of the animal world in terms of metabolic mediation [13]. The abundance of bacteria inside the tissues of a plant under normal conditions can reach even 10 million cells per gram of fresh tissue. On average, stems and roots of most plant species harbor a range from 10^3^ to 10^6^ live internal bacteria per gram of fresh weight, whose roles are related to different interactive phenotypes [14]. The effects on the physiology and on the responses that a plant can perform towards overall environmental stimuli are in essence highly influenced by the presence, abundance, and diversity of the endophytic microbial component [15]. Endophytes can have direct beneficial effects towards their host plant. They can accelerate seedling emergence, promote plant establishment under adverse conditions, improve tolerance to drought, and enhance plant growth. Endophytic microbes can foster productivity by helping plants in acquiring nutrients, e.g., via nitrogen fixation, iron chelation, phosphate solubilization, by preventing pathogen infections via antifungal or antibacterial agents, by outcompeting pathogens for nutrients via siderophore production, or by stimulating the plant’s systemic resistance. With regard to the power to detect and retrieve the vast array of taxa that inhabit plants, the key critical issues have been identified as the protocol for sample preservation and extraction, and the choice of primer sets for culture-independent DNA amplification [16]. The current knowledge on the endophytic microbiomes related to plant health has been recently reviewed [17].

Our group has been directly involved in several studies on microbial endophytes in the past decades. Direct DNA amplification-based detection of bacteria within legume roots at strain level has been improved by designing PCR primers based on unique insertion elements [18]. It was subsequently shown that several non-symbiotic co-occupants of root nodules occur in wild plants along with the nitrogen-fixing primary endophyte [19]. Later it was also demonstrated that plants can be the unexpected reservoir of bacterial species of human clinical relevance exploiting endophytically such alternative niche in their cycles [20], and that those plant-hosted mammalian pathogens could bear determinants of virulence towards mammals [21]. Protocols for endophyte visualization under epifluorescence microscopy have been optimized [22], and a state of non-culturability was observed to affect many endophytic bacteria and even symbionts within root nodules, calling for protocols that could counteract such impaired condition and restore their culturability on plates [23].

As the presence of bacterial endophytes, as well as their co-presence with symbionts in plants, entails profound consequences in hosts ecology and conservation outlook, in this report on a special issue devoted to the flora in the Mediterranean basin, we decided to focus on the legumes occurring on the island Sardinia and to review the status of our knowledge in terms of their associated microbiology. The survey involved screening existing literature about the nearly three hundred legume species of the Sardinian flora checklist and seeking reports in which authors would have determined the systematic identity of either nitrogen-fixing symbionts occurring in nodules, or that of other endophytic bacteria associated with nodules or found in other internal plant portions.

Additionally, we present a practical tool to determine plant taxonomy of field specimens using a system of subtractive keys by using the filter function in Microsoft Excel.

## 2. Results and Discussion

### 2.1. Plants Dataset Selection

The list of legume plant hosts was taken from Italian flora records [4] and verified for updates in the online database available at http://luirig.altervista.org/flora/taxa/floraindice.php (Accessed on 28 April 2022), from which the view by region option (Sardegna) and that of the family of choice (Fabaceae) were selected. Upon inspecting the genera, each resulting species and subspecies were cross-checked for synonyms and resolved to generate the list present in Appendix A, Appendix A. Sardinian Fabaceae checklist.xlsx. The records thereby featured amount currently to 290 taxa, classified as inclusive of either species or their subspecies when present.

### 2.2. Legumes Featuring Reports with Bacterial Taxonomy Characterization

Upon screening publications in Google Scholar search engine and using each plant taxon name (testing in parallel its synonyms) as keywords, plus the possible terms related to microsymbionts as different rhizobia or endophytes in the string, a vast series of publications was checked. The retrieved records were intended as those from overall geographical locations where those legumes would possibly occur, thus not restricted to the Sardinia island, where reports would have been much fewer to support a robust quantitative analysis. The data present in the published reports allowed us to select a subset of publications in which the identity of bacteria present in nodules or elsewhere in the plant internal portions (properly treated with surface sterilization) had been ascertained by adequate methods. These were considered DNA-based techniques, such as the small subunit ribosomal gene (16S) sequencing, and alignment with deposited records in the NCBI GenBank database (https://www.ncbi.nlm.nih.gov/ accessed on 30 May 2022), or results of DNA–DNA hybridization with cloned probes, or immunological detection by specific antisera. To consider bacteria detected in nodules as being the bona fide nitrogen-fixing symbionts and, at the same time, the putative inducer of the nodule formation, the experiments performed by the authors to demonstrate it were carefully evaluated. Those could also include reinfection tests in sterile microcosm conditions using surface-sterilized seeds of the original host legume, challenged with strains isolated from nodules of the plants. The proficiency in reinducing nodule formation and occupation was taken as evidence that the true endosymbiont had been identified. For those cases in which the authors themselves had defined their isolates as simply nodule-associated co-infecting endophytes or for those not isolated from nodules but from the internal portions of other plant organs, the endophyte attribution was accepted and separately listed in our elaborations.

Considering collectively the possible occurrences of either (a) nodule-inducing symbiont; (b) nodule-inhabiting endophyte; or (c) other plant parts endophyte, the subset of qualifying literature records yielded as a result a list of 105 species of Sardinian legumes that have been demonstrated to contain bacteria belonging to either one or more of those three categories. The list of plant genera and species and the number of cases in which they appear in the list is shown in Table 1. In this elaboration, the number of entries corresponding to a plant taxon is the sum of its signaled bacterial taxa, which could stem either from the same literature report, when different strains were found in plants collected from the same campaign, or from different independent literature records.

The most represented genus is *Medicago*, from which 50 occurrences are found. This situation could be partly expected also by the fact that the genus is highly featured in the flora of Sardinia in which it is present with 24 different taxa (see Appendix A Appendix A. Sardinian Fabaceae checklist.xlsx). The same applies for the *Trifolium* genus (present in Sardinia with 41 taxa), that also yielded 50 cases of associated bacteria, and for the *Vicia* genus (32 taxa in Sardinia) with 30 cases. The dominance of Medicago and Trifolium is, in fact, split across several plant taxa of them, which are in this sense more proportional to the local biodiversity of those two main genera within the Sardinian flora, also irrespective of the status of crops. The situation for Vicia shows instead only four cases, each with just one microbial instance.

However, the abundance of records in literature can depend also on the extent by which a legume is present worldwide, by its status of cropped plant, or just as present in spontaneous flora, and by the intensity of studies that deal with that plant, which are function of these mentioned variables. For this reason, for example, genera such as *Phaseolus* or *Pisum*, which are represented by just single species in Sardinia, are instead highly cited in overall literature and are here featuring 43 and 31 reported bacteria, respectively. It is worth noticing that some legumes, such as the *Sulla* genus, that includes three species in Sardinia (named under *Hedysarum* in the prior nomenclature), are a particularly rich source of data, due to a conspicuous number of studies that have dealt with them [18,19,20,21,22].

Another point to remark is that, among the resulting plants, all but one genus belong to subfamily Faboideae; the only exception being *Ceratonia*, member of the subfamily Caesalpinioideae. It appears that the other subfamilies might be understudied with respect to their microbial interactions of symbionts or endophytes.

### 2.3. Bacterial Symbionts in Nodules Reported for Legumes Included in the Sardinia List

In order to provide the complete body of results for the list of legumes that contained qualified descriptions of associated symbionts and endophytes, the compiled dataset reporting (a) symbionts, (b) endophytes in nodules, and (c) endophytes elsewhere in the plant is available as spreadsheet in Supplementary Material (Appendix A. Bacterial symbionts and endophytes by plant.xslx). The number of bacterial occurrences that resulted from the analysis of the symbionts amounts to 241 cases within 85 different taxa. Links to the corresponding literature reports accessible via web from the journal’s pages are included in the spreadsheet. Extracting the most relevant information from these data, a series of tables is elaborated below to analyze the observed situation, starting from the most recurring genus within the bona fide assigned bacteria which were declared as responsible for the host nodulation. In these studies, it is also to be considered that the actual taxon responsible for the nodule formation and subsequent nitrogen fixation could be in an unculturable state [19], calling for caution in assigning symbiotic roles that could lead to misassigned symbiotic inferences [24], and requiring culture-independent techniques such as direct PCR amplification from nodule tissue [19] or treatments with antioxidants to increase or recover culturability of bacterial cells [22].

Results at bacterial genus and species level of the 241 cases found, including a total of 74 species within 13 genera, are compiled in Table 2.

The widespread *Rhizobium* genus results as the top scoring one, followed by the *Mesorhizobium*, by the slow-growing *Bradyrhizobium* and by *Sinorhizibium*. Between these four genera and the rest there is a wide gap dividing the score in two groups with very different occurrence rate.

The identity of these bacteria is for most cases in line with common knowledge on nodulation proficiency, with some exceptions. In general, data allow to outline how the dominance of the *Rhizobium* genus can be in part attributed to a certain degree of unresolved taxonomy depth, as shown by the fact that the top entry is *Rhizobium* sp. and in part by the high level of diversification of described species within the genus, most of which, however, are single instances of detection and are found at the bottom of the score. *Mesorhizobium* is also confirmed as a highly diverse genus with several featured species.

It is also worth pointing out that some of the widely recurring symbionts (*S. meliloti* and *R. leguminosarum*) may be highly represented because they are symbiotic with species-rich host genera: *Medicago*, *Trifolium*, and *Vicia*.

Another comment to mention is that some of the counts may be affected by nomenclatural changes in bacterial taxonomy. As an example, before 2016, *Mesorhizobium japonicum* strains were placed in *Mesorhizobium loti*. Therefore, some of the reports of *Mesorhizobium loti* that were published before 2016 could be *Mesorhizobium japonicum*. The same situation could apply to some other recently described rhizobial species.

The cases that instead can be regarded as uncommon concern, for example, the finding of *Agrobacterium* in the list. Although it is a member of the Rhizobiaceae family, its presence in nodules is, by most authors, reported as endophytic; in this case, we maintained the report as the authors affirm to have verified that the isolate contains a copy of the *nodA* gene and explain their findings by hypothesizing a prior lateral gene transfer from rhizobia to *Agrobacterium* [25]. Another unconventional case is that of *Phyllobacterium myrsinacearum*, for which authors indicate to have verified symbiotic proficiency by nodulation tests [26]. A somewhat more puzzling case is the claim that *Pseudomonas* and *Burkholderia* could nodulate (the host was *Robinia pseudoacacia*), for which cases authors affirm the successful re-nodulation tests and the presence of nodulation and nitrogen-fixation genetic determinants [27]. In addition, the claim that *Paenibacillus* could nodulate *Trifolium pratense* [28] is again an unconventional finding that could deserve a confirm by further tests. In addition, it can be commented that both *Pseudomonas* and *Burkholderia* are known to be possible plant endophytes; therefore, finding them in root nodules may not be surprising as they are featured in other reports listed in the subsequent Table 3; besides, those studies are from 2010 and 2013, respectively, and there have not been any reports since then to confirm those findings.

### 2.4. Bacterial Endophytes in Nodules Reported for Legumes Included in the Sardinia List

The endophytes list appears at the same time the most rich in diversity, amounting to 142 different taxa, and the most evenly distributed of the lists commented so far, since the most abundant case, *Pseudomonas,* is found in only 16 cases, and the overwhelming majority of the other taxa are encountered twice or once. The recurring endophytes are represented by the classes and Gammaproteobacterial (*Pseudomonas*, *Enterobacter, Pantoea*) and the Gram-positive phylum of Firmicutes (*Bacillus*, *Paenibacillus*). For the rest, a peculiar case is the variety of different species belonging to the genus *Micromonospora* (Phylum Actinobacteria), all found in *Pisum sativum* nodules that were described in five different reports reviewed in a comprehensive survey chapter [29].

One aspect worth remarking is that on different occasions, bacteria which belong to genera or species which are also known to be symbiotic (such as the *Rhizobium* and *Mesorhizobium* genera) are not considered the primary symbionts but just other endophytes. We have verified this situation directly when assessing the recovery of culturability from nodules in which there were different taxa of the Rhizobiaceae; only one of them was recognized as the true symbiont [22]. In this respect, the rhizobia can be defined as bacteria that are particularly proficient in endophytism, some of which have refined and deepened their plant interaction by establishing a full mutualism.

### 2.5. Bacterial Endophytes Reported as Occurring in Other Plant Portions for Legumes Included in the Sardinia List

In a number of reports, although less frequent, authors also investigated the identities of endophytic bacteria in non-nodular tissues such as other root portions, stems, or leaves. These microbial taxa amount to 33 different ones, the most recurring of which are within *Paenibacillus* sp. and *Arthrobacter* sp., with three and two occurrences, respectively. Overall, the members of this community tend to be similar to those of the nodule endophytes, suggesting that an entry into nodules, facilitated by a co-invasion with the infecting rhizobial symbiont, and a possible migration from the nodule to other plant compartments could be part of a common pathway. The data are summarized in Table 4.

One of the plants from which endophytes were also sought in extra-nodular tissues assumes a particular relevance for the present paper as it is about a Sardinian endemism, namely, the legume *Astragalus terraccianoi*, that was moreover studied along with a plant of the Asteraceae, *Centaurea horrida*, with whom the legume is part of the phytosociological association *Centaureetum horridae* Mol. [22]. In that investigation we analyzed both species in their typical habitat, finding them along the windswept cliffs on the rocky shores of Asinara, a small uninhabited island over the northwestern coast of Sardinia, which for a long time was only used as location of a prison. That report deals with bacteria belonging to all three categories that we are using for this review: nodule symbiont, nodule endophytes, and other portions endophytes, which in that case were analyzed from the same plant specimens. Densities of endophytes ranged between 3.7 × 10^2^ and 2.8 × 10^4^ colony forming units per gram of plant fresh weight. The identities of non-nodular root and stem endophytes, ascertained by 16S rDNA sequencing, included *Actinobacterium* sp., *Paenibacillus* sp., *Rhizobium* sp., *Methylobacterium* sp., *Pedobacter panaciterrae, Aerococcus viridans,* and *Stenotrophomonas rhizophila.* The putative nodule symbiont instead, described thereby for the first time, was in a non-culturable state, as is common for several Mediterranean spontaneous legumes [19], and required PCR amplicon sequencing for its identification, which pointed out at a 97% sequence similarity with *Bradyrhizobium canariense.* It was also found to co-inhabit nodules with several different endophytes, such as *Bacillus sporothermodurans, Bacillus pumilus, Bacillus simplex, Bacillus flexus, Streptomyces ciscaucasicus,* and *Dyella* sp. [22].

### 2.6. Most Promiscuous Plants in Terms of Symbionts or Endophytes Content

Upon cross-analyzing the above tables, it was possible to extract complementary information, i.e., which are the plants that, comparing them across different reports, result to be hosting the highest diversity of possible nodule symbionts or nodule endophytic taxa. Data are shown in Table 5. The cases of stem or root endophytes are not included, as the number of reports featuring those was much lower in comparison to the other two categories.

The microsymbiont promiscuity indicated a broad range (= low host specificity) for *Phaseolus vulgaris* and *Vigna unguiculata*, in which the variety of possible symbionts was equated by a corresponding richness in endophytic types, but also in *Biserrula pelecinus* and *Cicer arietinum*, in which cases, on the contrary, the endophytes reported were none or few, respectively. On the opposite scale, some legumes showed a narrow symbiont range but a vast potential to contain different endophytes, such as the case of *Astragalus terraccianoi*, *Hedysarum spinosissimum (Sulla spinosissima)*, *Hedysarum glomeratum (Sulla capitata)*, and *Medicago sativa*.

### 2.7. Most Promiscuous Bacteria in Terms of Plant Nodulation or Endophytic Infection

The complementary type of data regards the most successful bacteria in terms of symbiotic plant host range nodulation or as endophyte invader. The data are shown in Table 6.

It can be seen that *Sinorhizobium meliloti*, which is the nitrogen-fixing partner of alfalfa, fenugreek, and *Melilotus*, is at the top of the score for symbionts; the datum is due to the wide variety of cropped or spontaneous *Medicago* species that are featured in the reports. *Rhizobium leguminoasarum* is next, and its abundance is also due to the existence of biovars (bv. *viciae*, bv. *phaseoli*, bv. *trifolii*) that make this species the typical symbiont of peas, lentils, beans, and clovers.

With regard to the recurring endophytes, taxa such as the Gram-positive (Firmicutes) of the *Bacillus* group dominate, followed by the Gammaproteobacteria of the Enterobacteriaceae family (*Erwinia*, *Pantoea*), which are all known to be proficient in endophytism also in plant families different from the Fabaceae.

### 2.8. BOTABASE KEYS: An Interactive File and Customizable Model Tool for Plant Determination

Finally, we wished to include in this report, and make it available for the community of botanists and microbiologists, a practical tool that we created as a Microsoft Office Excel file, and which we have successfully used during our campaigns for the Sardinian legume analyses as a practical aid to plant species determination either directly in the field or on collected specimens. Since our expeditions and studies were carried out before the publication of the novel edition of the Italian flora, which was completed in 2019 [4], the file covers the Sardinian legume species that were included in the prior edition of Pignatti’s Flora d’Italia botanical guide [30], in which the taxa of Fabaceae for Sardinia were 189. However, the file can be implemented at leisure by just including the new taxa as new rows and adding into the existing columns their data, gathered from the updated Italian flora botanical guide [4,5]. Moreover, the concept can serve as a basis to make corresponding files for families different from the Fabaceae, as well as for those of other regions and countries for which the basic morphology and distribution data would be available.

For this tool, the following files are available in Supplementary Material:

Appendix A. English version Botabase Keys for Legumes of Sardinia Island.xls.

Appendix A. Versione italiana Botabase Keys per Leguminose Sardegna.xls.

Document S1. English version Instructions for Botabase Keys.doc.

Document S2. Istruzioni versione Italiana per Botabase Keys.doc.

In essence, the tool is meant as a user-friendly on-site botanical identification system; it was devised on the basis of the following premises:(1)It does not necessarily require a professional knowledge in botany and can be managed by users across different levels of education, upon becoming familiar with some anatomical words for which, if necessary, they can consult a glossary of botanical terms, among the several freely available web pages that can be found by search engines.(2)The end user will just require a personal computer, in which the process runs as a simple Microsoft Excel spreadsheet. This offers the added advantage of most users being already familiar with the required interface. Unlike other tools using answers for narrowing down the number of species in biological determinations, this one does not require a dedicated software.(3)The plant identification at species level is achieved in few sequential and rapid steps that filter off progressively larger groups of plants from the initial full database and can lead to a single entry even before the grid of answers is completed, or at least to few possibilities, out of which the correct one can be, in most cases, easily chosen by looking at plant pictures from the web upon searching the plant species name in Google search engine and asking for images. As many biometric data are used, very often the level of information contained in the spreadsheet for each single plant is overly redundant, and classification to the single species can be achieved even by answering only half of the questions or less.(4)The user works by *subtractive keys*. The identification principle is parallel rather than serial, i.e., one can proceed even in the absence of some of the data (e.g., the fruit is not available or the plant has not even flowered yet). Questions of uncertain answer can be skipped at any step without preventing the process from reaching completion. Such a feature is an advantage when compared to standard methods of classification which follow a single path of binary keys, where a missing element can stop the process or a wrong answer at any fork can lead to a wrong identification.(5)The keys are designed in a such a way that enables them to be “flexible and forgiving”, which matches one of the inherent qualities of any biological array: the plasticity and variability of life. The process avoids the most commonly occurring determination errors encountered with other methods by applying two practices: (a) choosing “loose borders” around values, i.e., using thresholds that extend above and below the confidence boundaries for any of the observable variables (thus avoiding excluding false negatives), and (b) enabling to skip questions on which the answer is uncertain or not possible due to a missing element (e.g., the fruit). These non-restrictive principles are compensated by the multiple questions of the whole procedure that ensure a sharp final accuracy level.(6)The database of plants can be easily implemented whenever new species should be found in a range (adding new rows), or new convenient traits for the determination need to be added (adding new columns), or any data need to be corrected or updated. Such features can even be performed by the end user, leading to a full customization of the tool.(7)Visual satisfaction and transparent appreciation of the procedure accompanies the identification. Unlike other query-based computer methods in which the procedure is blind, the user of Botabase Keys starts with a spreadsheet in which the full array of species is visible in the rows. Then, proceeding across columns and answering the questions by applying the simple filter function of Microsoft Excel, the list is trimmed down at each step (seeing the table being cropped over and over and watching the rows disappear at every further click of the mouse is, moreover, a pleasing game-like effect that makes the process rather entertaining, which is envisaged as a way to attract people to the study of botany). The possibility of skipping columns of uncertain answer or starting from any point of the path eliminates the possible frustration that occurs in classical dichotomy-based error-prone methods.(8)The program is free to use and share and users are encouraged to implement it, update it, correct possible errors, and customize it to suit their needs in the study of different plants families, regions, and parts of the world, or by creating an upper key to family determination as well as any improvement that they can envisage.

## 3. Materials and Methods

### 3.1. Data Collection and Elaboration

The Google Scholar search engine was used to retrieve literature. The search string involved the plant taxon name (genus and species) and the terms rhizob* or endophyt*, which were used separately in search rounds. Records were inspected by reading titles and abstracts; articles corresponding to the required description of bacterial taxa were accessed and downloaded. When methods of taxa assignment corresponded to reliable standards (16S rDNA gene sequencing and database alignment, DNA–DNA hybridization, or immunological techniques) and adequate surface sterilization protocols to avoid contamination from external biota, data were gathered and used to compile a worksheet table with the following distinction: (1) bacteria assigned by the authors as true nitrogen-fixing symbionts (in which case primary sequencing data were backed up by the authors’ descriptions of axenic in vitro host nodulation tests using strains isolated from nodules); (2) bacteria considered nodule-associated endophytes (co-occurring in nodules along with the primary symbiont); (3) bacteria occurring elsewhere inside plant tissues (other than root nodules). Scores of abundances of cases per plant host and of co-occurrence were elaborated to generate the tables presented as a result of this survey. Since after the articles publication some plant hosts underwent nomenclatural emendations and changes, the original plant names used by the authors of the records are reported in the results table, and correspondence to updated botanical nomenclature can be verified by inspecting the list present in Appendix A: Appendix A. Sardinian Fabaceae checklist.xlsx, in which synonyms related to former nomenclature are reported in brackets aside the current taxon denominations.

### 3.2. Plant Determination File (BOTABASE KEYS)

The method falls under the “Design of question contents” definition. It makes use of (a) a database of plants in which their data (available in floristic treatises and guides) have been accurately entered and (b) a method to interrogate the database by answering a number of questions on easily observable aspects. In the present case, it was constructed to suit the needs of identification of the Leguminosae species that can be found within the Sardinia region. The program is compiled in two versions, English and Italian, and is available in Appendix A. English version Botabase Keys for Legumes of Sardinia Island.xls or Appendix A. Versione italiana Botabase Keys per Leguminose Sardegna.xls.

The Microsoft Excel spreadsheet has plants listed horizontally (in the rows) and their features vertically (in the columns). The procedure leading to identification is based on a text-filtration process upon observing the plant features and entering the corresponding values in the Excel filter mask. The progressive use of the filter enables one to reduce, step by step, the list of rows.

The goal is to crop out all the rows except the one corresponding to the specimen under examination. In essence, the 31 columns allow to interrogate the matrix and filter away the rows that do not correspond to the specimen’s observable features. In case of missing information, columns can be simply skipped to proceed with the subsequent ones. Thus, unlike the case of dichotomy keys, the process would not be stuck nor mislead to a wrong branch. The traits on which the plants are checked out are the following: minimum altitude, maximum altitude, minimum height, maximum height, flowering since (month), flowering until (month), leaf shape, minimum number of leaflets, maximum number of leaflets, leaflets shape, leaflet minimum width, leaflets maximum width, leaflets minimum length, leaflets maximum length, inflorescence type, minimum n. of flowers per inflorescence, maximum n. of flowers per inflorescence, minimum length of flowerhead, maximum length of flowerhead, corolla color, minimum length of corolla, maximum length of corolla, legume type, other notes, habitat, habitat notes, substrate, abundance, endemism, and geographical notes. A detailed series of instructions in English and in Italian are included in the Supplementary Material: Document S1. English version Instructions for Botabase Keys.doc” or Document S2. Istruzioni versione Italiana per Botabase Keys.doc”.

## Figures and Tables

**Table 1 plants-11-01521-t001:** Number of occurrences in which genera and species of legumes belonging to the Sardinia checklist have been studied in terms of their bacterial symbionts or endophytes associated microbiology that led to taxonomical assignments. Plant names correspond to the current botanical nomenclature (http://luirig.altervista.org/flora/taxa/floraindice.php accessed on 30 May 2022). The original plant names used by the authors of the records, for plants whose names have changed, can be matched by inspecting taxa synonyms in Appendix A: Appendix A. Sardinian Fabaceae checklist.xlsx.

Species	n	Species	n	Species	n
** *Anagyris* **	**4**	** *Lens* **	**4**	** *Pisum* **	**31**
*Anagyris foetida*	4	*Lens culinaris*	4	*Pisum sativum*	31
** *Astragalus* **	**32**	** *Lotus* **	**53**	** *Robinia* **	**19**
*Astragalus boeticus*	1	*Lotus angustissimus*	1	*Robinia pseudoacacia*	19
*Astragalus hamosus*	1	*Lotus conimbricensis*	1		
*Astragalus pelecinus*	16	*Lotus corniculatus*	21	** *Scorpiurus* **	**12**
*Astragalus terraccianoi*	14	*Lotus cytisoides*	1	*Scorpiurus muricatus*	11
		*Lotus edulis*	1	*Scorpiurus vermiculatus*	1
** *Bituminaria* **	**4**	*Lotus maritimus*	1		
*Bituminaria bituminosa*	4	*Lotus ornithopodioides*	1	** *Spartium* **	**3**
		*Lotus parviflorus*	11	*Spartium junceum*	3
** *Ceratonia* **	**3**	*Lotus subbiflorus*	1		
*Ceratonia siliqua*	3	*Lotus tenuis*	3	** *Sulla* **	**33**
		*Lotus tetragonolobus*	10	*Sulla capitata*	10
** *Cicer* **	**18**	*Lotus uliginosus*	1	*Sulla coronaria*	7
*Cicer arietinum*	18			*Sulla spinosissima*	16
		** *Lupinus* **	**14**		
** *Colutea* **	**8**	*Lupinus luteus*	1	** *Trifolium* **	**50**
*Colutea arborescens*	8	*Lupinus albus*	3	*Trifolium campestre*	1
		*Lupinus angustifolius*	4	*Trifolium diffusum*	1
** *Coronilla* **	**1**	*Lupinus micranthus*	6	*Trifolium dubium*	1
*Coronilla valentina*	1			*Trifolium fragiferum*	4
		** *Medicago* **	**50**	*Trifolium nigrescens*	1
** *Cytisus* **	**31**	*Medicago arabica*	1	*Trifolium ornithopodioides*	1
*Cytisus laniger*	9	*Medicago ciliaris*	1	*Trifolium pratense*	22
*Cytisus scoparius*	16	*Medicago doliata*	1	*Trifolium repens*	13
*Cytisus spinosus*	3	*Medicago hispida*	7	*Trifolium strictum*	1
*Cytisus villosus*	3	*Medicago intertexta*	1	*Trifolium suffocatum*	1
		*Medicago litoralis*	1	*Trifolium tomentosum*	4
** *Dorycnium* **	**1**	*Medicago lupulina*	1		
*Dorycnium hirsutum*	1	*Medicago murex*	1	** *Trigonella* **	**11**
		*Medicago orbicularis*	3	*Trigonella elegans*	1
** *Ervilia* **	**9**	*Medicago praecox*	1	*Trigonella marítima*	1
*Ervilia hirsuta*	9	*Medicago rigidula*	1	*Trigonella monspeliaca*	1
		*Medicago rugosa*	2	*Trigonella officinalis*	1
** *Ervum* **	**3**	*Medicago sativa*	21	*Trigonella sicula*	1
*Ervum tetraspermum*	3	*Medicago scutellata*	1	*Trigonella smalii*	6
		*Medicago tenoreana*	1		
**Genista**	**1**	*Medicago tornata*	1	** *Vicia* **	**30**
*Genista monspessulana*	1	*Medicago truncatula*	4	*Vicia disperma*	1
		*Medicago turbinata*	1	*Vicia faba*	19
** *Glycyrrhiza* **	**6**			*Vicia lathyroides*	2
*Glycyrrhiza glabra*	6	** *Melilotus* **	**1**	*Vicia leucantha*	1
		*Melilotus italicus*	1	*Vicia nigricans*	2
** *Hippocrepis* **	**6**			*Vicia peregrina*	1
*Hippocrepis multisiliquosa*	2	** *Ononis* **	**4**	*Vicia sativa*	1
*Hippocrepis unisiliquosa*	4	*Ononis natrix*	1	*Vicia sepium*	2
		*Ononis ornithopodioides*	1	*Vicia villosa*	1
** *Hymenocarpos* **	**1**	*Ononis spinosa*	2		
*Hymenocarpos circinnatus*	1			** *Vigna* **	**28**
		** *Ornithopus* **	**16**	*Vigna unguiculata*	28
** * Lathyrus * **	** 13 **	*Ornithopus compressus*	5		
*Lathyrus aphaca*	1	*Ornithopus perpusillus*	6		
*Lathyrus clymenum*	1	*Ornithopus pinnatus*	5		
*Lathyrus latifolius*	6				
*Lathyrus pratensis*	5	** *Phaseolus* **	**43**		
		*Phaseolus vulgaris*	43		

**Table 2 plants-11-01521-t002:** Number of occurrences for the given bacterial genera and for their encompassed species, detected in nodules of legumes which are featured in the Fabaceae of Sardinia, and identified by the authors of the reports as the actual symbiont.

Species	n	Species	n	Species	n
** *Rhizobium* **	**82**	** *Mesorhizobium* **	**62**	** *Bradyrhizobium* **	**45**
*R. leguminosarum*	21	*Mesorhizobium sp.*	16	*Bradyrhizobium sp.*	17
*Rhizobium sp.*	20	*M. loti*	10	*B. canariense*	8
*R. etli*	4	*M. ciceri*	4	*B. japonicum*	7
*R. laguerreae*	4	*M. huakuii*	4	*B. elkanii*	4
*R. anhuiense*	3	*M. chacoense*	3	*B. cytisi*	2
*R. phaseoli*	3	*M. mediterraneum*	3	*B. liaoningense*	2
*R. pisi*	3	*M. tianshanense*	3	*B. lupini*	2
*R. acidisoli*	2	*M. amorphae*	2	*B. rifense*	2
*R. gallicum*	3	*M. japonicum*	2	*B. yuanmingense*	1
*R. hidalgonense*	2	*M. temperatum*	2		
*R. indigoferae*	2	*M. abyssinicae*	1	** *Sinorhizobium* **	**32**
*R. sophorae*	2	*M. albiziae*	1	*S. meliloti*	20
*R. sullae*	2	*M. australicum*	1	*S. medicae*	5
*R. aethiopicum*	1	*M. erdmani*	1	*Sinorhizobium sp.*	4
*R. cellulosilyticum*	1	*M. intechi*	1	*S. fredii*	3
*R. chutanense*	1	*M. jarvisii*	1		
*R. indicum*	1	*M. muleiense*	1	** *Neorhizobium* **	**6**
*R. leucaenae*	1	*M. opportunistum*	1	*N. galegae*	2
*R. lusitanum*	1	*M. plurifarium*	1	*N. huautlense*	2
*R. mesosinicum*	1	*M. robiniae*	1	*Neorhizobium sp.*	2
*R. multihospitium*	1	*M. shonense*	1		
*R. rhizogenes*	1	*M. thiogangeticum*	1	** *Phyllobacterium* **	**3**
*R. ruizarguesonis*	1	*M. wenxinie*		*P. myrsinacearum*	2
*R. tropici*	1			*Phyllobacterium sp.*	1
*R. vallis*	1	** *Agrobacterium* **	**2**		
		*Agrobacterium sp.*	1	** *Burkholderia* **	**1**
** *Ensifer* **	**2**	*A. tumefaciens*	1	*Burkholderia sp.*	1
*Ensifer sp.*	2				
		** *Paenibacillus* **	**1**		
** *Microvirga* **	**2**	*Paenibacillus sp.*	1		
*Microvirga sp.*	2				
		** *Pseudomonas* **	**1**		
** *Pararhizobium* **	**2**	*Pseudomonas sp.*	1		
*P. giardinii*	2				

**Table 3 plants-11-01521-t003:** Number of cases reported as endophytes co-occurring within the legume nodules but not considered responsible for nodule formation nor necessarily in symbiotic relationship with the host.

Species	n.	Species	n.	Species	n.
*Pseudomonas sp.*	16	*Bacillus circulans*	1	*Micromonospora ureilytica*	1
*Bacillus sp.*	12	*Bacillus flexus*	1	*Micromonospora vinacea*	1
*Paenibacillus sp.*	10	*Bacillus insolitus*	1	*Mucilaginibacter sp.*	1
*Bacillus megaterium*	9	*Bacillus kochii*	1	*Mycobacterium sp.*	1
*Enterobacter sp.*	7	*Bacillus mojavensis*	1	*Novosphingobium sp.*	1
*Bacillus simplex*	6	*Bacillus pumilus*	1	*Ochrobactrum ciceri*	1
*Mesorhizobium sp.*	6	*Bacillus sporothermodurans*	1	*Ochrobactrum sp.*	1
*Phyllobacterium sp.*	6	*Bordetella avium*	1	*Oerskovia sp.*	1
*Erwinia persicina*	4	*Bosea sp.*	1	*Ornithinicoccus sp.*	1
*Pantoea ananatis*	4	*Brevibacillus agris*	1	*Paenibacillus sp.*	1
*Streptomyces sp.*	4	*Burkholderia sp.*	1	*Paenibacillus endophyticum*	1
*Acinetobacter sp.*	3	*Buttiauxella sp.*	1	*Paenibacillus kribbensis*	1
*Agrobacterium sp.*	3	*Caulobacter sp.*	1	*Paenibacillus lupini*	1
*Agrobacterium tumefaciens*	3	*Chitinophaga sp.*	1	*Paenibacillus polymixa*	1
*Ancylobacter sp.*	3	*Chryseobacterium sp.*	1	*Paraburkholderia nodosa*	1
*Enterobacter agglomerans*	3	*Cohnella lupini*	1	*Paracoccus sp.*	1
*Sphingomonas sp.*	3	*Cupriavidus sp.*	1	*Phyllobacterium endophyticum*	1
*Staphylococcus pasteuri*	3	*Curtobacterium citreum*	1	*Phyllobacterium ifriquiensis*	1
*Xanthomonas sp.*	3	*Curtobacterium flaccumfaciens*	1	*Phyllobacterium loti*	1
*Achromobacter sp.*	2	*Curtobacterium luteum*	1	*P. myrsinacearum*	1
*Arthrobacter sp.*	2	*Delftia sp.*	1	*Promicromonospora sp.*	1
*Bacillus subtilis*	2	*Enterobacter cloacae*	1	*Providencia sp.*	1
*Bacillus thuringiensis*	2	*Fontibacillus phaseoli*	1	*Pseudomonas brassicacearum*	1
*Brevibacillus sp.*	2	*Herbaspirillum lusitanum*	1	*Pseudomonas brenneri*	1
*Corynebacterium sp.*	2	*Kaistia sp.*	1	*Pseudomonas corrugata*	1
*Dyella sp.*	2	*Klebsiella sp.*	1	*Pseudomonas frederiksbergensis*	1
*Herbaspirillum sp.*	2	*Luteibacter sp.*	1	*Pseudomonas putida*	1
*Inquilinus sp.*	2	*Lysobacter sp.*	1	*Pseudomonas rhodesiae*	1
*Kocuria sp.*	2	*Massilia sp.*	1	*Pseudomonas yamanorum*	1
*Leifsonia sp.*	2	*Micromonospora aurantiaca*	1	*Rahnella aquatilis*	1
*Lysinibacillus sp.*	2	*Micromonospora carbonacea*	1	*Rahnella sp.*	1
*Micromonospora lupini*	2	*Micromonospora chokoriensis*	1	*Ralstonia pickettii*	1
*Micromonospora saelicesensis*	2	*Micromonospora coxiensis*	1	*Rhizobium hidalgonense*	1
*Pantoea agglomerans*	2	*Micromonospora halophytica*	1	*Rhizobium radiobacter*	1
*Pantoea sp.*	2	*Micromonospora humi*	1	*Rhizobium vignae*	1
*Pseudomonas fluorescens*	2	*Micromonospora krabiensis*	1	*Rhodococcus sp.*	1
*Pseudomonas fragi*	2	*Micromonospora luteifusca*	1	*Serratia liquefaciens*	1
*Rhizobium leguminosarum*	2	*Micromonospora luteiviridis*	1	*Serratia plymuthica*	1
*Rhizobium nepotum*	2	*Micromonospora marina*	1	*Serratia proteamaculans*	1
*Rhizobium sp.*	2	*M. matsumotoense*	1	*Starkeya novella*	1
*Sphingobacterium sp.*	2	*Micromonospora mirobrigensis*	1	*Stenotrophomonas sp.*	1
*Staphylococcus epidermidis*	2	*Micromonospora noduli*	1	*Stenotrophomonas maltophilia*	1
*Staphylococcus sp.*	2	*Micromonospora phytophila*	1	*Streptomyces sp.*	1
*Stenotrophomonas sp.*	2	*Micromonospora pisi*	1	*Streptomyces ciscaucasicus*	1
*Variovorax sp.*	2	*M.purpureochromogenes*	1	*Thiobacillus sp.*	1
*Actinoplanes sp.*	1	*Micromonospora rifamycinica*	1	*Variovorax paradoxus*	1
*Agrobacterium rhizogenes*	1	*Micromonospora siamesi*	1		
*Bacillus brevis*	1	*Micromonospora sp.*	1		

**Table 4 plants-11-01521-t004:** Number of cases of endophytes reported as having been isolated from plant tissues other than nodules from plants belonging to the Sardinian checklist.

Species	n.	Species	n.
*Paenibacillus sp.*	3	*Novosphingobium sp.*	1
*Arthrobacter sp.*	2	*Paenibacillus enshidis*	1
*Bacillus sp.*	2	*Pantoea agglomerans*	1
*Actinobacterium sp.*	1	*Pantoea sp.*	1
*Aerococcus viridans*	1	*Pedobacter panaciterrae*	1
*Agrobacterium sp.*	1	*Pseudomonas sp.*	1
*Bosea robiniae*	1	*Rahnella sp.*	1
*Chryseobacterium sp.*	1	*Rhizobium sp.*	1
*Curtobacterium sp.*	1	*Shinella sp.*	1
*Endobacter medicaginis*	1	*Sinorhizobium sp.*	1
*Herbaspirillum robiniae*	1	*Stenotrophomonas rhizophila*	1
*Klebsiella sp.*	1	*Stenotrophomonas sp.*	1
*Leifsonia sp.*	1	*Streptomyces sp.*	1
*Methylibium sp.*	1	*Tardiphaga robiniae*	1
*Methylobacterium sp.*	1	*Variovorax sp.*	1
*Micromonospora sp.*	1	*Xanthomonas sp.*	1
*Mycobacterium sp.*	1		

**Table 5 plants-11-01521-t005:** List of the Sardinian legumes featuring the highest numbers of occurrence for taxonomically different bacteria, either as symbionts or as nodule endophytes. Plants are listed in alphabetical order.

Plant	N. of # Symbiont Taxa	N. of # Endophyte Taxa
*Astragalus terraccianoi*	2	12
*Biserrula pelecinus*	16	0
*Cicer arietinum*	13	5
*Cytisus scoparius*	5	10
*Hedysarum glomeratum*	2	7
*Hedysarum spinosissimum*	2	14
*Lotus corniculatus*	2	19
*Lotus parviflorus*	1	10
*Medicago sativa*	3	15
*Phaseolus vulgaris*	15	21
*Pisum sativum*	7	24
*Robinia pseudoacacia*	8	11
*Trifolium pratense*	8	14
*Vicia faba*	9	10
*Vigna unguiculata*	10	13

**Table 6 plants-11-01521-t006:** List of bacteria that resulted as particularly widespread across the different plant taxa either as symbionts (left side of the table) or as nodule endophytes (right side of the table).

Symbionts	n. of # Plants	Endophytes in Nodules	n. of # Plants
*Sinorhizobium meliloti*	20	*Bacillus megaterium*	9
*Rhizobium leguminosarum*	16	*Bacillus simplex*	6
*Mesorhizobium loti*	9	*Erwinia persicina*	4
*Bradyrhizobium japonicum*	7	*Pantoea ananatis*	4
*Bradyrhizobium canariense*	6	*Agrobacterium tumefaciens*	3

## Data Availability

All data used in this research are available in the Appendix A.

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
