# Peer review of "Legumes of the Sardinia Island: Knowledge on Symbiotic and Endophytic Bacteria and Interactive Software Tool for Plant Species Determination"

_plants, 2022, doi:10.3390/plants11111521_

Round 1

Reviewer 1 Report

I think that the subject of the manuscript is interesting and ready for publication.

Reviewer 2 Report

The present article presents a meta-analysis carried out on records of Fabaceae plants and their reported symbionts/endophytes, and a tool for plant identification.

In its present form, the manuscript presents several weak points and flaws that should be addressed by the authors:

1) the article doesn't draw any conclusions, nor seems to start from a specific research question. After reading it several times, I still do not clearly understand what is the underlying scientific question that is behind the meta-analysis that the authors carried out, nor what is the link between the meta-analysis and the species identification tool.

2) while the authors do state sufficiently their reasons to include many records from outside Sardinia in their meta-analysis, they should be careful in their analysis of the plant-microbe association when comparing data that come from very different soils in very different geographic areas. Number of reports and type of association with the plant are strongly influenced by what kind of bacteria are present in the soil that is examined, therefore what happens in the plants found in Sardinia could be very different from what the literature presents.

3) Table 1 could be expanded to include the number of reports of bacterial association beside the number of plants per genus.
Likewise, Table 3 could give data regarding the three levels of association identified by the authors (nodule-inducing, nodule-associated, endophyte), rather than just a single number condensating all these.

Reviewer 3 Report

Please see the attached .pdf file.

Round 2

Reviewer 2 Report

The authors' replies cleared that the article is providing the intended content, requested by the editors for this special issue, and that the manuscript is meant to be a review. Therefore, I have no further comments on the organization and content of the manuscript.

The changes to the tables are satisfactory and make the content easier to read.